# Swelling Effects on the Conductivity of Graphene/PSS/PAH Composites

**DOI:** 10.3390/nano11123280

**Published:** 2021-12-03

**Authors:** Tianbao Zhao, Ruyi Yang, Zhi Yang

**Affiliations:** 1School of Materials Science and Engineering, Xihua University, Chengdu 610039, China; zhaotb@mail.xhu.edu.cn (T.Z.); Laurance_young@mail.xhu.edu.cn (R.Y.); 2National Research Center of Pumps, Jiangsu University, ZhenJiang 212013, China

**Keywords:** graphene, polyelectrolyte, swelling, sheet resistance, molecular dynamics simulation

## Abstract

Graphene/poly-(sodium-4-styrene sulfonate)(PSS)/poly-(allylamine hydrochloride) (PAH) composite is a frequently adopted system for fabricating polyelectrolyte multilayer films. Swelling is the bottleneck limiting its applications, and its effects on the conductivity is still controversial. Herein, we report successful swelling of a graphene/PSS/PAH composite in a vapor atmosphere, and the relation with the mass fraction of water is uncovered. The composite was prepared via a layer-by-layer assembly technique and systematically characterized. The results indicated that the average thickness for each bilayer was about 0.95 nm. The hardness and modulus were 2.5 ± 0.2 and 68 ± 5 GPa, respectively, and both were independent of thickness. The sheet resistance decreased slightly with the prolongation of immersion time, but was distinct from that of the water mass fraction. It reduced from 2.44 × 10^5^ to 2.34 × 10^5^ ohm/sq, and the change accelerated as the water mass fraction rose, especially when it was larger than 5%. This could be attributing to the lubrication effect of the water molecules, which sped up the migration of charged groups in the polyelectrolytes. Moreover, molecular dynamics simulations confirmed that a microphase separation occurred when the fraction reached an extreme value owing to the dominated interaction between PSS and PAH. These results provide support for the structural stability of this composite material and its applications in devices.

## 1. Introduction

Polyelectrolyte multilayer (PEM) films have been extensively explored in the last decade due to their high chemical and physical stability and their high permeability and selectivity. They have shown great promise in applications such as fabricating ultrathin conducting layers, permeation-selective membranes, protective coatings, sensors, and materials for nonlinear optics [1]. For example, poly-(ethylene)-based delivery systems, such as polyelectrolyte composites (PECs), poly-(ethylene)-coated nanocarriers, and poly-(ethylene) (PE) multilayers, were designed to protect peptides and proteins from degradation and facilitate their absorption. These delivery systems are especially effective when administered orally or intranasally [2]. The K^+^/Li^+^ selectivity can reach ∼60 through poly(allylamine hydrochloride)/poly-(sodium-4-styrene sulfonate)-coated Nafion membranes, which are much larger than the uncoated Nafion membranes with a K^+^/Li^+^ selectivity of <3. This because the swelling of poly(allylamine hydrochloride)/poly-(sodium-4-styrene sulfonate) (PAH/PSS) multilayers may open cation-exchange sites that preferentially bind K^+^ to enable highly selective transport [3]. In addition, PPE-SO^3−^ molecules exhibit interesting optoelectronic properties, making graphene-based materials potentially useful in a variety of optoelectronic device applications [4].

The PAH/PSS was the first system where the stratified nature of linearly growing multilayers was established, and now, it is a widely used polyelectrolyte combination for the buildup of PEM films [5]. PSS and PAH are charged in ionic solvents and exhibit a self-assembly ability due to their electrostatic interaction, viscoelasticity, and hydrophilic action. Based on layer-by-layer (LBL) assembly technology, a nano-scale composite membrane with excellent structure can be prepared by changing the polyelectrolyte and controlling the deposition conditions, such as the concentrations of PAH and PSS, deposition time, pH, number of layers, etc. Feldötö used quartz crystal microbalance (QCM) and dual polarization interferometry to illuminate the structure of PAH/PSS multilayer films in two different salts, KBr and NaCl [6]. Because bromide ions are larger and have greater polarizability and, thus, stronger interactions with PAH, which leads to weaker intrachain repulsion, PAH adopts a more coiled structure in solution and, therefore, produces thicker layers. The PAH/PSS multilayer contains a very compact, rigid core with a small amount of water incorporated into the film, independently of the electrolyte and rinsing procedure. When rinsed with pure water, only the outermost layer swells (to a thickness of 25−30 nm), and the underlying layers retain their compact structure. Relaxation measurements showed that the outermost layer and the water content influenced the mobility of the polyelectrolytes and the adsorbed water [7]. Unexpectedly, PAH/PSS multilayers were resistant to NaCl with a concentration up to 1.2 M. Using (infrared active) perchlorate salt, the fraction of residual counterions in PAH/PSS was determined to be 6%. The free energy of association between the polymer segments in the presence of NaClO_4_ was about −10 kJ mol^−1^ for PAH/PSS, indicating the relatively strong association between the polymer segments [8]. Varying the pH of the solution in contact with the PAH/PSS multilayer revealed a transition to a highly swollen state, which was interpreted to signal protonation of PAH under much more basic conditions than the pKa of the solution polymer. The increase in the multilayer pKa suggested an interaction energy for PAH/PSS in NaCl of ca. 16 kJ mol^−1^ [8]. Furthermore, the size or volume fraction of the nanoparticle in the nanocomposite also affected the crystallization properties [9,10] and has been observed in various nanocomposites [11,12].

Despite extensive investigations that revealed the factors affecting the building of PEM films, relatively little is known about the humidity that exists within these applications and the methods that control their swelling. For PAH/PSS assembly films, swelling in water was deteriorative for the stability of the nanostructures. Graphene oxide and its derivatives were composited to overcome this obstacle. For example, homogeneous and controllable -SO_3_Na and graphene multilayer films formed via π–π interactions not only dispersed well in water owing to the ionic side chains of -SO_3_Na, but also prevented the aggregation of graphene by electrostatic repulsion [13]. In this paper, we focused on the effects of humidity on the swelling of graphene/PSS/PAH composite films with a layer-by-layer assembly technique, in which polyelectrolytes and graphene with different charges adsorbed alternatively in solution and arbitrarily shaped functional coatings could be prepared at nano-scale [14]. The primary strategies involve exfoliation of graphene oxide into poly-(sodium-4-styrene sulfonate) (PSS) solution with the aim of stabilizing graphene oxide sheets, followed by in situ reductions to produce PSS-stabilized reduced graphene oxide (PSS-rGO) sheets. Then, positively charged PAH and negatively charged PSS-rGO were alternately adsorbed on substrate until the hybridized functional film grew up to the 100th bilayer. The obtained films were swollen in water vapor and their sheet resistances were tested. The relationship between the water mass fraction and sheet resistance was investigated, and the mechanisms were uncovered through theoretical calculations.

## 2. Experimental Section

### 2.1. Materials 

Natural graphene flake (Grade-300), poly-(sodium-4-styrene sulfonate) (PSS, M_w_ = 70,000), poly-(ethylenimine) (PEI, 50% in H_2_O, M_n_ = 60,000 by GPC, M_w_ = 750,000 by LS), and poly-(allylamine hydrochloride) (PAH, M_w_ = 50,000–65,000) were all purchased from Sigma-Aldrich (Tianjin, China). Concentrated sulfuric acid (H_2_SO_4_, 95%), sodium nitrate (NaNO_3_), hydrochloric acid (HCl), hydrazine monohydrate (N_2_H_4_·H_2_O), hydrogen peroxide (H_2_O_2_), potassium permanganate (KMnO_4_), and other chemicals were supplied by Sinopharm Chemical Reagent CO., Ltd. (Tianjin, China). All chemicals and reagents were used directly as received. Deionized water (DI, >18 MΩ) was produced with a Milli-Q Plus system (Millipore) (Tianjin, China).

### 2.2. Preparation of PSS-Stabilized Graphene Sheets (rGO-PSS)

Graphene sheets are usually subject to large van der Waals forces, which cause them to stack together. One of the major challenges in fabricating them is to disperse the graphene sheets without using covalent chemistries or other harsh conditions, which could lower their electrical conductivity. To achieve this purpose, graphene oxide sheets with negatively charged carboxylate groups were re-dispersed in PSS (1.00 mg/mL, pH = 6.15) solution. The π-like stacking of benzene rings of PSS benefits the binding of PSS and graphene sheets, leading to a high fraction of individualized graphene after dispersion and reduction. Details: Graphene oxide was synthesized by Hummer’s method [15]. Graphene oxide (0.1250 g) was firstly dissolved in ultrapure water (250 mL), followed by ultrasound treatment at 500 W of power for 50 min. After vacuum suction filtration, the filtrate was taken as liquid sample A. PSS (2.000 g) was dissolved in ultrapure water (250 mL), followed by addition of 2 mL of 0.5 mol/L sodium hydroxide solution drop by drop to adjust the solution pH to 8, and the obtained solution was taken as liquid sample B. Liquid sample A was poured into liquid sample B under stirring, and then filtrated with vacuum suction filtration. Finally, hydrazine hydrate (500 μL) was added into the filtrate, followed by stirring and reflux for 24 h at 90 °C, and then cooled to room temperature.

### 2.3. Preparing (rGO-PSS/PAH)_n_ Multilayer Films (n: Number of Deposition Cycles)

Reduced graphene oxide composite was assembled on glass following the schematic shown in Figure 1a. All supporters were ultrasonically cleaned in absolute ethyl alcohol, propone, and isopropyl alcohol in sequence, and then cleaned with a mixed solution of H_2_SO_4_/H_2_O_2_ (*v*/*v*: 3/1) at 80 °C for 1 h [16]. Generally, multilayer deposited on surfaces was not thermodynamically favored, and so it seemed unavoidable, as strong interlayer mixing and chain removal, and would be triggered upon exposure to a deposition solution containing one of the polyelectrolytes [17]. Hence, the substrate was preferred to be firstly immersed into PEI (1.00 mg/mL, pH = 7.25) solution for 10 min, followed by rinsing with DI water to generate a cationic surface, which ensured that the surface charge was uniformly decorated to improve the adhesion strength. Since the binding of polyelectrolytes on the substrate surface was prior to the dissolution step, removal of the previously adsorbed polymer chain was inhibited when the pure buffer was used instead of the polymer solution at the early stage of desorption. Subsequently, a rinsing process was conducted to follow each cycle with an alternative dipping, rinsing, and drying procedure with the PAH (1.00 mg/mL, pH = 7.26) and PSS-rGO (0.5 mg/mL) solutions. Additional bilayers were added using identical procedures. For each cycle, a bilayer of PAH/PSS–graphene was prepared; the UV–visible spectra of the growing bilayers were recorded in air after each five assembly cycles. The inset in Figure 1b shows the self-supporting 100-bilayer-thick assembled film transferred to a glass sheet after being peeled off as a result of hydrogen fluoride corrosion. 

### 2.4. Swelling and Sheet Resistance Tests

The traditional method, which involves immersing the assembled film in electrolyte solution for a certain amount of time, followed by washing and drying, leads, no doubt, to mass loss, especially for the polyelectrolytes in the film. Disruption of inter-polyelectrolyte binding also lies in the heart of salt-induced and pH-induced dissolution of multilayers. So as to avoid an erosion phenomenon in pure water, we swelled the composite film with water vapor at room temperature. The prepared composite films were immersed in water vapor for 24 h to complete the swelling, and then were dried in the air to reach a target mass (or target H_2_O mass fraction). After that, the sheet resistance was measured using a four-probe method.

### 2.5. Molecular Dynamics Simulations 

The cuboid simulation cells were sized 14.76 Å × 14.76 Å × 25.00 Å. There were four monomer units for PSS and PAH in each cell. The element compositions and densities are listed in Table 1. The volume fraction of the reduced graphene oxide sheet was about 7.46%. All dynamics simulations (MDs) were performed using a COMPASS force field. Both energy and geometrical optimization were performed, and a spherical cutoff value of 12.5 Å was chosen to calculate the non-bond interactions. All calculations were carried out at 300 K in an NPT ensemble, which was controlled by Andersen’s method. Periodic boundary conditions along all three directions were used. Based on the convergence of potential, kinetic, non-bond, and total energy, the MD simulation ran for 1000 ps to relax and reach an equilibrium state with a time step of 1 fs.

### 2.6. Characterizations

#### 2.6.1. Spectrum Analysis

The spectroscopic properties were investigated using the UV-Vis spectra (Hitachi U-4100 UV-Vis near-FTIR spectrometer, Hitachi, Tokyo, Japan) with a scanning range of 200–800 nm and a scanning speed of 300 nm·min^−1^. Raman spectra (Renishaw micro-Raman 1000 spectrometers, Renishaw, British, Gloucestershire, UK) and FT-IR spectra (Fourier-Transform Infrared Spectrometer 3100) were assessed under a laser wavelength of 532 nm and an intensity of 1.5 mW. The wave number range was 4000–400 cm^−1^. A minimum of 16 scans signal-averaged with a resolution of 2 cm^−1^ was employed.

#### 2.6.2. Morphology Characterizations 

The morphologies were investigated with an atomic force microscope (AFM, Bruker, Karlsruhe, Germany) on a Veeco Icon in the taping mode, and scanning electron microscopy (SEM, FEI, Hillsboro, OR, USA,) was performed on a HITACHI S4300 system at the accelerated voltage of 20 kV. Transmission electron microscopy measurements were performed on a Tecnai G2 F30 microscope (Thermo Scientific, Eindhoven, Netherlands) at 80 kV.

#### 2.6.3. Mechanical Tests 

The mechanical properties were tested by using nano-indentation experiments on a Nano Indenter XP system after a standard calibration. The Poisson ratio was set to 0.25, the surface-approach velocity was 10 nm/s, the strain rate target was 0.05/s, and the frequency target was 45 Hz. 

#### 2.6.4. Sheet Resistance Evaluations 

Keithley S4200 (Tektronix, Beaverton, WA, USA) was used to measure the sheet resistances with a four-probe method under different levels of humidity at room temperature (30 °C). 

## 3. Results and Discussion

### 3.1. Characterizations of PSS-Stabilized Graphene Sheets (rGO-PSSs)

Figure 2a shows practical photos of the PSS-stabilized graphene oxide (GO-PSS) solution and rGO-PSS solution. Because of the changes in the light absorbance properties, the color of the solution became black after a chemical reduction. The small aggregations at the bottom indicate that the reduced graphene oxide sheets were stabilized by PSS molecules. The rGO-PSS sheet had a lamellar structure with a size of tens of microns according to the optical microscope view in Figure 2b and the AFM images in Figure 2c. Figure 2d shows a low-magnification TEM image of typical rGO-PSS sheets over a copper grid. They exhibited typical wrinkle-like features, which were in good agreement with previous reports [18].

### 3.2. Fabrication of Polyelectrolyte/rGO-PSS Composite via the Layer-by-Layer Assembly Method

Polycations with a high density of primary amino groups and polyanions with SO_3_^−^ groups showed the strongest inter-polyelectrolyte binding, resulting in inhibited chain exchange within PEMs. Prior to the assembly, PEI was first absorbed on the substrate to enhance the binding strength between the film and substrate. The growth of the multilayers was monitored using the UV-Vis spectrum. The UV-Vis curves are shown in Figure 3. The characteristic absorption peak appearing at 225 nm was attributed to the π-π* transitions of aromatic C=C bonds. A progressive increase in absorbance in the range of 200–800 nm was visible when the bilayer number increased. The inset shows that, in the early stage of assembly, the linear relationship between the absorbance and the bilayer numbers indicated a reproducible and uniform LBL assembly process, which provided a feasible scheme for controlling the thickness of the film at the nanometer level. The polyelectrolyte multilayers of PAH and PSS grown from ion-free solution showed a large surface charge density of 2–3 mC m^−2^ (absolute values). Note that PAH and PSS had the same linear charge density; thus, the surface coverage was the same, and each adsorption step was accompanied by charge overcompensation [18]. During the multilayer preparation, at 60 bilayers, a transition from a linear to a parabolic growth regime occurred. In the linear growth regime, the surface of the film was thought to be electrically neutral. Only upon the decrease in the NaCl concentration below a critical value (50 mM in the literature) was a surface charge observed. At lower salt concentrations, Cl^−^ ions dissolved from the multilayer, which was now positively charged. The surface charge density was again unusually low at 0.62 ± 0.22 mC m^−2^. Actually, the uniformity of the composite was key in obtaining excellent mechanical properties.

### 3.3. Raman and FT-IR Spectrum Analysis

Figure 4a shows the transmittance FT-IR spectra of the GO-PSS, rGO-PSS, and composite film. The stretching vibrations of the –OH group were located in the range of 3000 to 3700 cm^−1^. The decrease in the peak intensity indicated that most of the –OH groups in graphene oxide were removed after the reduction by N_2_H_4_·H_2_O. The peaks of the C=O group were located between 1650 and 1750 cm^−1^ owing to asymmetric stretching. The peaks at 1540 and 1182 cm^−1^ were assigned to C=C bonds and C–O–C groups. The peak at 800 cm^−1^ was attributed to para-disubstituted benzene rings. The Raman peaks located at 1580 and 1380 cm^−1^ were G and D bands, respectively [19,20]. The intensity of the D band enhanced the red shift over five units after reduction, revealing the hybridization type transformation of the carbon skeletons: sp^3^ to sp^2^.

### 3.4. Morphology Characterizations 

A typical AFM test in the taping mode was performed, which indicated an average film thickness of 95 nm on a SiO_2_/Si substrate (Figure 5a,b). The optical microscope photograph in Figure 5c shows darker spots on the surface of the film, which were aggregated fine particles. The surface morphologies of the 100 bilayers were characterized by SEM, as shown in Figure 5d–f. The SEM images indicate a smooth surface with large numbers of wrinkles at a small scale. In addition, the obvious layered characteristics of the interface cross-section were present. 

### 3.5. Modulus and Hardness 

The mechanical properties, e.g., elastic modulus and hardness, were characterized by a nano−indentation method owing to its excellent performance on at the nano−scale. The variations in the hardness and modulus with the indentation depth are shown in Figure 6a, and those with the film thickness are shown in Figure 6b. This demonstrated that the average hardness and modulus were about 2.5 ± 0.2 and 68 ± 5 GPa for the 100−bilayer film, which made the composite film easy to peel off and transfer to an arbitrary target. Furthermore, the relationships between the stiffness and modulus of multilayers with different thicknesses are exhibited in Figure 6b. The stiffness and modulus of the multilayers increased slightly with the thickness, but not significantly.

### 3.6. Sheet Resistances and Theoretical Calculations

The effect of the swelling of thin films on the sheet resistance is the key for further consideration. Figure 7a shows the variations in the curves of sheet resistance with the immersion time. They decreased slightly from 2.45 × 10^5^ to 2.38 × 10^5^ ohm/sq after immersion for 24 h. The sheet resistance decreased slightly with the prolongation of immersion time; however, the modified value was less than only 10% and was eventually saturated, which should not affect the fundamental properties. The influence of the H_2_O mass fraction, however, was different. It was reduced from 2.44 × 10^5^ to 2.34 × 10^5^ ohm/sq, and the rate of change increased with the increase in the H_2_O mass fraction, especially when the mass fraction was larger than 5%. This can be attributed to the lubrication effect of the water overflow in the film, which accelerated the migration of charged groups into the polyelectrolytes. However, when the mass fraction of H_2_O continued to increase, the interaction between PSS and PAH dominated and a microphase separation occurred, which led to restacking of the graphene sheets and, thus, a decrease in sheet resistance. Furthermore, graphene sheets could help to stabilize the polyelectrolyte films and greatly reduce the inner resistance by changing the transport of the electrons from a “point-to-point” mode to a more effective “plane-to-point’’ mode [21]. 

Figure 8 shows the atomic configurations of the graphene/PSS/PAH composites with different mass fractions of water. Under anhydrous condition, the interaction between PSS and graphene based on π-π bonding and that between PSS and PAH was mainly electrostatic. These non-bond interactions were often weak, so the composite structures could remain stable. However, when the mass fraction of water increased, water molecules were free between the polyelectrolyte molecules and played a special role in lubricating and connecting PSS and PAH molecules through hydrogen bonds and intermolecular interactions. In theory, mixing like−charged macromolecules before compositing should offer a significant scope for controlling the composition and properties of polyelectrolyte composites/coacervates [22]. Figure 9 shows the total energy and its contributions to the graphene/PSS/PAH composites with different water mass fractions. Remarkably, the valence energy changed little with the vibration of the water mass fraction, which was distinct from the vibration of the non-bond energy. When the mass fraction was zero, the non-bond energy was positive, which means that the system was metastable. When the mass fraction was larger than 5%, the non-bond energy became negative. When the water content reached a certain degree, the interaction between PAH and PSS was greater than that between PSS and graphene, and microphase separation occurred. The graphene layers were more properly restacked, and hence, the conductivity was improved.

## 4. Conclusions

In summary, we report the successful fabrication of a graphene/PSS/PAH composite for swelling studies. The composite was prepared via a layer-by-layer assembly technique and systematically characterized by UV-Vis, Raman, and FT-IR spectra, atomic force microscopy, a nano-indentation procedure, a four-probe method, and a theoretical calculation. An average film thickness of 0.95 nm was found for each bilayer. The average hardness and modulus were about 2.5 ± 0.2 and 68 ± 5 GPa, and both were independent of the thickness, which made the composite film easy to peel off and transfer to an arbitrary target. The sheet resistance decreased slightly with the prolongation of the immersion time. The influence of the H_2_O mass fraction, however, was different. It was reduced from 2.44 × 10^5^ to 2.34 × 10^5^ ohm/sq, and the rate of change increased with the increase in the H_2_O mass fraction, especially when the mass fraction was larger than 5%. This could be attributed to the lubrication effect of the water overflow in the film, which accelerated the migration of charged groups into the polyelectrolytes. When the mass fraction of H_2_O continued to increase, the interaction between PSS and PAH dominated and a microphase separation occurred. These results provide support for applications of the graphene/PSS/PAH thin films. 

## Figures and Tables

**Figure 1 nanomaterials-11-03280-f001:**
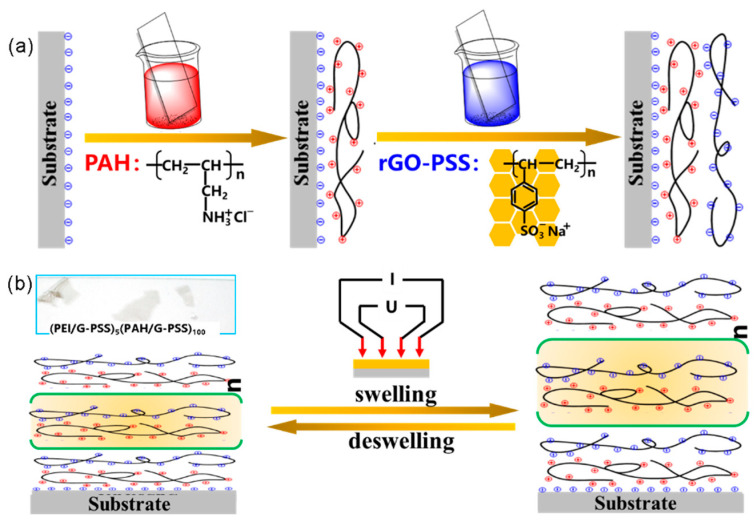
(**a**) Schematic diagram of one layer-by-layer assembly cycle. The black solid line represents the changed and reduced graphene oxide. The symbols “⊖” and “⊕” represent the negative and positive charges, respectively. The red-colored solution is PAH and the blue-colored solution is rGO-PSS. (**b**) The sheet resistance affected by H_2_O concentrations was tested with a four-probe method.

**Figure 2 nanomaterials-11-03280-f002:**
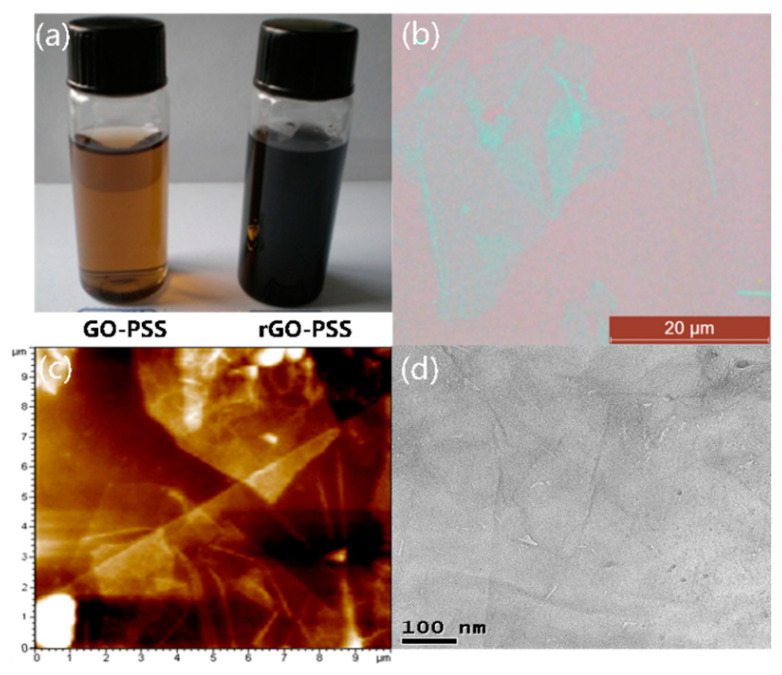
Characterizations: (**a**) GO-PSS solutions before and after reduction; (**b**) optical micrograph of GO-PSS sheets; (**c**,**d**) AFM and TEM images of rGO-PSS sheets, respectively.

**Figure 3 nanomaterials-11-03280-f003:**
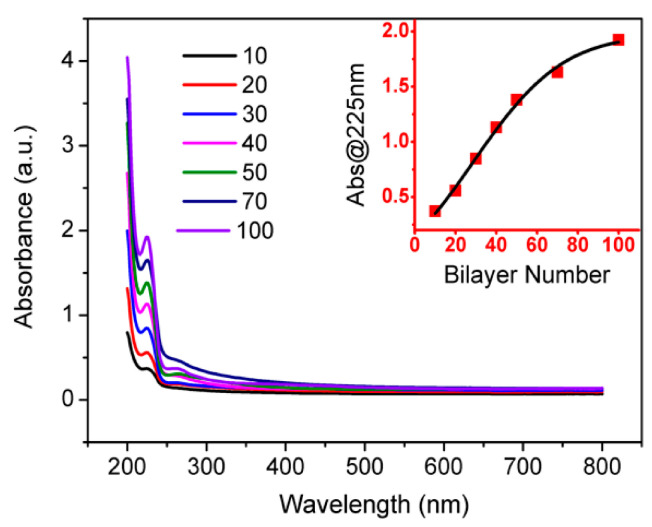
UV-Vis spectra of the (PAH/graphene–PSS)_100_ multilayer film growing on a quartz slide. The inset shows the relationship of absorbance at 225 nm versus the number of bilayers assembled.

**Figure 4 nanomaterials-11-03280-f004:**
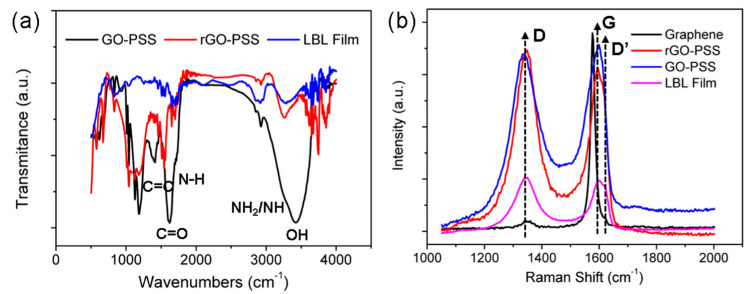
(**a**) FT−IR and (**b**) Raman spectra of GO−PSS, rGO−PSS, and the 100−bilayer LBL film.

**Figure 5 nanomaterials-11-03280-f005:**
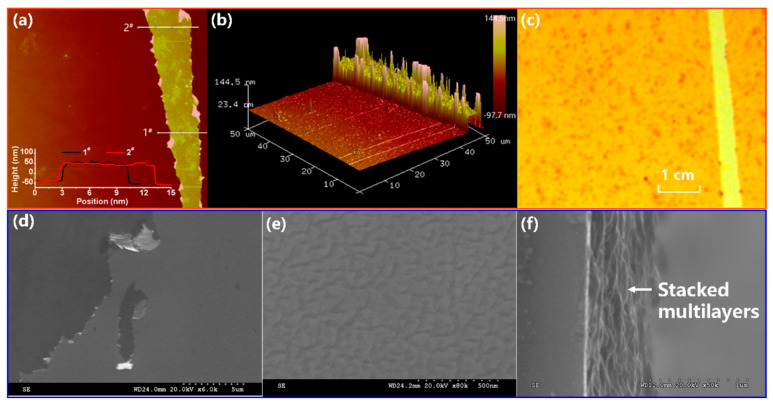
AFM images (**a**,**b**), optical micrograph image (**c**), and SEM images (**d**–**f**) of the (PAH/rGO−PSS)_100_(PEI/PSS)_2_ film.

**Figure 6 nanomaterials-11-03280-f006:**
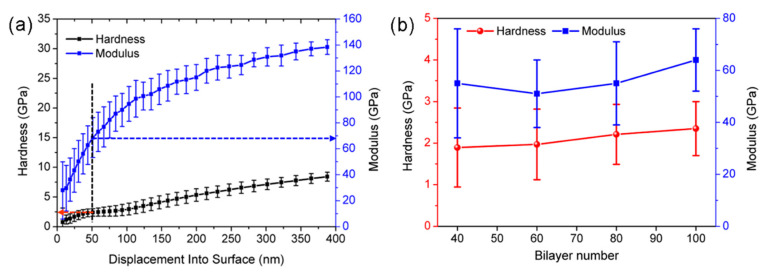
(**a**) Nano-indentation tests of the assembled 100−bilayer film; (**b**) variation in the hardness and modulus with thickness.

**Figure 7 nanomaterials-11-03280-f007:**
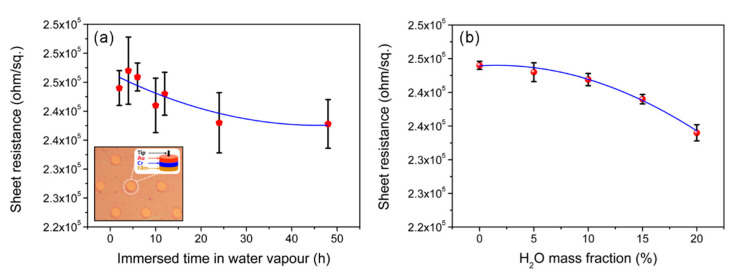
Sheet resistance variations with the swelling time (**a**) and the mass fraction of H_2_O (**b**).

**Figure 8 nanomaterials-11-03280-f008:**
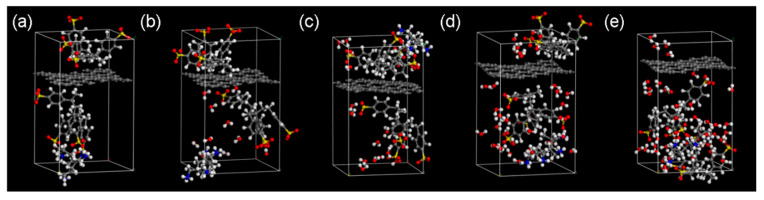
Relaxed atomic structures of the graphene/PSS/PAH composites with water mass fractions of (**a**) 0%, (**b**) 5%, (**c**) 10%, (**d**) 15%, and (**e**) 20%. The oxygen (red), sulfur (yellow), hydrogen (white), nitrogen (blue), and carbon (gray) atoms were distinctively colored.

**Figure 9 nanomaterials-11-03280-f009:**
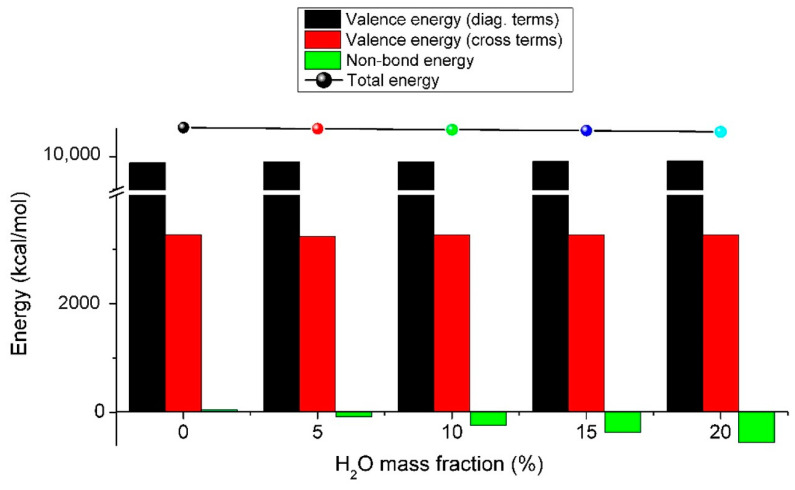
Total energy and contributions of graphene/PSS/PAH composites with different water mass fractions.

**Table 1 nanomaterials-11-03280-t001:** Elemental compositions of the five simulation models. The numbers represent the number of atoms of each element in all simulation models.

Percentage of H_2_O /%	C	H	O	N	S	H_2_O	Density/g·cm^−3^
0	212	94	24	4	8	0	0.783
5	212	103	33	4	8	9	0.838
10	212	132	43	4	8	19	0.888
15	212	156	55	4	8	31	0.53
20	212	178	68	4	8	44	1.025

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
