# Peer review of "Swelling Effects on the Conductivity of Graphene/PSS/PAH Composites"

_nanomaterials, 2021, doi:10.3390/nano11123280_

Round 1

Reviewer 1 Report

In this manuscript, the authors report the study of polyelectrolyte multilayer graphene nanocomposite systems and the effect of swelling on their conductivity. In addition, they have studied the effect of water content on the electrical properties of the nanocomposite. They have also done some molecular dynamics simulations. Finally, they have also measured the mechanical properties and thickness of individual layers in the multilayer structure. The work seems to be suitable for consideration in nanomaterials, subject to some modifications outlined below.

-Introduction

-Some of the abbreviations should be explained as soon as they are used. (e.g poly-(sodium-4-styrene sulfonate) for PSS)

- In the introductory statement, there is no mention of the MD simulation they have done in this work as part of their study. One only discovers this when reading page 4. The effect of nanocomposite concentration of different components affects their properties and has been observed in various nanocomposites. A few significant recent MD simulation works have explained the effect of nanoparticles size, shape, and volume fraction could be mentioned. [e.g Nanomaterials 9 (10), 1472; Nanoscale Adv., 2019,1, 4704-4721]. These molecular simulations have nicely explained some of the experimental observations [e.g. J. Chem. Phys., 2017, 147, 020901; Polymer, 2018, 157, 111–121.]. I suggest the authors mention some of the works above to show the broad nature of the phenomena they have studied in their work. This is also very relevant as they have done some MD simulations in this manuscript.

-Methods:

I believe they have used “reduced graphene oxide” (rGO). However, they have mentioned this much later after they use the term “rGO”.

  • Change the title of section 2.5 from “dynamic simulations” to “molecular dynamics simulations.”
  • The method for MD simulations is too brief. They could extend the description of the methods a little be more in detail. For example, what was the volume fraction of the rGO sheets in the system? Does it appear there is only a single rGo sheet? Is this a good representation of the hybrid system? Why is an NVT system used instead of NPT?

Results

While there are extensive experimental results, the simulation results only include snapshots and energy levels of systems with different water content. That would be good to include some of the structural information, if possible, from the simulations.

Presentations:

The work needs thorough proofreading to stem out the poor English and correct typographical errors.

Author Response

To Reviewer #1:

Q#1: Some of the abbreviations should be explained as soon as they are used. (e.g poly-(sodium-4-styrene sulfonate) for PSS); In the introductory statement, there is no mention of the MD simulation they have done in this work as part of their study.

ANSWER: All abbreviations were revised as suggested and the MD simulations results were added in the part of “Introduction”.

Q#2: The effect of nanocomposite concentration of different components affects their properties and has been observed in various nanocomposites. A few significant recent MD simulation works have explained the effect of nanoparticles size, shape, and volume fraction could be mentioned. [e.g Nanomaterials 9 (10), 1472; Nanoscale Adv., 2019,1, 4704-4721]. These molecular simulations have nicely explained some of the experimental observations [e.g. J. Chem. Phys., 2017, 147, 020901; Polymer, 2018, 157, 111–121.]. I suggest the authors mention some of the works above to show the broad nature of the phenomena they have studied in their work. This is also very relevant as they have done some MD simulations in this manuscript.

ANSWER: Thank you for your suggestions. The introduction has been revised carefully and literatures above-mentioned were cited and discussed appropriately.

Q#3: I believe they have used “reduced graphene oxide” (rGO). However, they have mentioned this much later after they use the term “rGO”.

ANSWER: The abbreviation and quotation of some professional terms were normalized, such as “rGO” mentioned later, were replaced by “reduced graphene oxide”.

Q#4: Change the title of section 2.5 from “dynamic simulations” to “molecular dynamics simulations.”

ANSWER: We have changed the title of section 2.5 from “dynamic simulations” to “molecular dynamics simulations”.

Q#5: The method for MD simulations is too brief. They could extend the description of the methods a little be more in detail. For example, what was the volume fraction of the rGO sheets in the system? Does it appear there is only a single rGO sheet? Is this a good representation of the hybrid system? Why is an NVT system used instead of NPT?

ANSWER: More simulation details were added. The volume fraction of the rGO sheets in the system was about 7.46%. For each cell, there is only a single rGO sheet, but periodic boundary condition were used along all three directions. Besides, the “NVT” in section 2.5 was changed to “NPT” because of a written mistake.

Q#6: While there are extensive experimental results, the simulation results only include snapshots and energy levels of systems with different water content. That would be good to include some of the structural information, if possible, from the simulations.

ANSWER: Thank you for your good suggestions. Frankly speaking, there is no need to supply more structural information to support the experimental results.

Q#7: The work needs thorough proofreading to stem out the poor English and correct typographical errors.

ANSWER: The English language of the manuscript and pictures have been checked and revised carefully.

Reviewer 2 Report

This is an outstanding piece of work. My only recommendations are stylistic. i.e. I'd write certain paragraphs differently, but the meaning would remain unchanged. As such I am trying a new thing, and instead not offering that feedback - this paper is perfectly fine and should be accepted without further review.

Author Response

Thank you very much for reviewing our manuscript carefully and giving us good suggestions. We have revised it according to the comments and resubmitted for further consideration.  

Reviewer 3 Report

the author report successful fabrication of graphene/PSS/PAH composite via layer-by-layer assembly technique, the sample was fully characterized by UV-Vis, Raman and FT-IR spectra, atomic force microscope, nano-indentation procedure, a four-probe method and a theoretical calculation. the relaionship between hardness and modulus withthe swelling of layers was investigated showing no dependecy on the thickness. The composite film resulyed to be easy to be peeled off and transferred to target. Similarly, the author investigated the relationship between sheet resistance and swelling due to water. 

the paper is well written and each aspect is fully analysed and explained, so I suggest this manuscript to be published in the present form.

Author Response

Thank you very much for reviewing our manuscript carefully and giving us good suggestions. We have revised it according to the comments and resubmitted for further consideration. All the changes are highlighted by red color in the text.